# Solidification Crack Evolution in High-Strength Steel Welding Using the Extended Finite Element Method

**DOI:** 10.3390/ma13020483

**Published:** 2020-01-19

**Authors:** Zhanglan Chen, Jianmin Liu, Haijun Qiu

**Affiliations:** 1School of Marine Engineering, Jimei University, Xiamen 361016, China; 200661000058@jmu.edu.cn; 2Institute of Welding Technology, Xiamen Shipbuilding Industry Co., Ltd., Xiamen 361021, China; qiuhaijun@163.com

**Keywords:** solidification cracking, 3-D modeling, fracture evolution, fracture mode

## Abstract

High-strength steel suffers from an increasing susceptibility to solidification cracking in welding due to increasing carbon equivalents. However, the cracking mechanism is not fully clear for a confidently completely crack-free welding process. To present a full, direct knowledge of fracture behavior in high-strength steel welding, a three-dimensional (3-D) modeling method is developed using the extended finite element method (XFEM). The XFEM model and fracture loads are linked with the full model and the output of the thermo-mechanical finite element method (TM-FEM), respectively. Solidification cracks in welds are predicted to initiate at the upper tip at the current cross-section, propagate upward to and then through the upper weld surface, thereby propagating the lower crack tip down to the bottom until the final failure. This behavior indicates that solidification cracking is preferred on the upper weld surface, which has higher weld stress introduced by thermal contraction and solidification shrinkage. The modeling results show good agreement with the solidification crack fractography and in situ observations. Further XFEM results show that the initial defects that exhibit higher susceptibility to solidification cracking are those that are vertical to the weld plate plane, open to the current cross-section and concentratedly distributed compared to tilted, closed and dispersedly distributed ones, respectively.

## 1. Introduction

High-strength steel (HSS) is widely used as a cost-effective, high load-carrying capacity, and light self-weight solution in modern shipbuilding and marine structure engineering. Alloying is a general process for HSS to achieve better tensile properties. However, higher alloy content exposes HSS to poor toughness due to its higher carbon equivalent. HSS thus has a higher susceptibility to solidification cracking in the welding process [1,2]. Since solidification cracking is an important consideration in HSS welding, understanding the fracture behavior of the solidification cracking in HSSs has, therefore, driven modeling efforts for decades.

Various modeling methods were employed to reveal the mechanical aspects and corresponding parameters were proposed to correlate with the mechanical behavior of solidification cracking. The thermo-mechanical finite element method (TM-FEM) has been proven to be accurate enough to detail the mechanical performance of solidification cracking in the welding process. With the output of TM-FEM, mechanical parameters such as stress and strain [3,4,5], equivalent plastic strain [6], and critical strain rate [7] were characterized as driving forces of solidification cracking. The phase field method (PFM) has also gained much popularity for modeling mechanical aspects of solidification cracking by focusing on the formation of weld microstructure during solidification. Based on the output of PFM, not only the tensile strain introduced by weld solidification shrinkage and thermal deformation [8,9], but also grain boundary [10], weld pool shape, and solidification morphology [11] have been proven to contribute to solidification cracking. Although numerical methods with crack-free models were shown to capture thermo-mechanical fields of welds with considerable precision [12], they can hardly give enough insight into the singularities introduced by metallurgy impurities or inadequate fluid feeding, since they are found to be the key initiators and promoters of solidification cracking.

With a three-dimensional (3-D) weld model with no thickness crack along the weld’s transverse direction, conventional fracture-modeling based on finite element modeling (FEM) was employed to simulate weld solidification cracking. The stress intensity factor (SIF)-based fracture parameter was introduced to investigate the solidification-cracking mode and to weigh the contribution of weld stress components [13]. This conventional fracture modeling accounted for the static interaction between the initial defect and weld matrix. The continuum damage mechanics model [14] and the Gurson–Tvergaard–Needleman material model [15] were put forward to predict the crack behavior of weld metal and the damage of structural components with macrostructure defects, respectively. However, for the full understanding of the solidification cracking mechanism, the understanding of the propagation evolution is still absent.

The extended-finite-element method (XFEM) gives an insight into the complex cracking evolution that leads to failure. Different from conventional fracture-modeling FEM schemes, XFEM defines discontinuities as enriched features without requiring the mesh to conform to the geometric discontinuities. Moreover, XFEM copes with the geometric discontinuity by meshing the crack surface, which facilitates the simulation of the evolution of a crack along a completely solution-dependent path. After it was first introduced by Belyschko and Black [16], XFEM has been extensively used to model various fracture problems. As far as weld fields are concerned, Qian and Zhao [17] utilized XFEM to assess the fatigue behavior of dissimilar welded steel joints.

Using the dynamic output data of XFEM, SIFs can be accessed readily. Zhou and Xue [18] utilized the displacement–extrapolation method for extracting SIFs. Moreover, The same method was employed to calculate the SIF by measuring the displacements at the near-tip nodes and crack length in a 3-D cracking problem [19,20]. 

The brittle nature of solidification cracking leads to the consideration of crack growth as a fracture problem that falls into the framework of linear elastic fracture mechanics (LEFM) [4,5,6,21]. Furthermore, XFEM has been proven to be effective in modeling brittle and ductile fractures and is, therefore, introduced here for the modeling of the solidification-cracking evolution. X-ray-based experimental investigation provides defect aspects of structure and mechanical behavior [22], and corresponding results were found for the validation of modeling ones. The motivation behind this investigation is to completely understand solidification cracking, including the fracture mode and fracture sequence, both of which can give an explicit insight into the mechanical behavior of the solidification cracking upon welding of HSS materials.

## 2. Materials and Methods

### 2.1. Experiment

#### 2.1.1. Material and Welding-Process Experiment

The material selected for the weld components was Q690D high-strength steel produced by a thermo-mechanical control process (TMCP). With nominal yield strength as high as 690 MPa, Q690D is typically used in ship and marine structures. Plates of 6 mm thickness were fabricated using Ar + 20% CO_2_ hybrid-gas-shielded welding. A butt weld with a dimension of 50 mm × 300 mm, a single V-shaped groove with an interval of 4 mm, and an inclination angle of 53° was configured. The weld components were restrained following the standard method of the rigid restraint cracking test for the welding butt joint (GB/T 13817-92), which is a typical method for testing a material’s hot cracking sensibility. The other parameters were: Current = 190–230 A, arc voltage = 25–28 V, welding speed = 400 mm/min, filler metal MK.ER80-G (satisfying both the standard requirements of AWS A5.28M:2005 (American welding society) and GB/T 8110-2008) with a diameter of Φ1.2 mm (series 18-3243), gas flow = 15–20 L/min, and preheating temperatures of 130–150 °C.

Moreover, welding temperature was measured simultaneously. On the upper surfaces of the weld components, four points—crossing at longitudinal sections with distances of 20 and 30 mm from the longitudinal centerline, and transverse sections with distances of 175 and 250 mm from the weld’s starting point—were selected as measurement points. After the weldment cooled down, the residual stresses were measured using the blind-hole stress-release technology by means of DH5922 dynamic strain equipment.

#### 2.1.2. Uniaxial Tensile Test and Measurement

After welding, the weld’s chemical constitution was measured, as shown in Table 1.

Five sets of specimens for uniaxial tensile tests were abstracted from butt–joint in terms of tensile test methods on the weld and deposited metal (GB/T 2652-2008/ISO 5178: 2001). Each set has three specimens for the consideration of taking an average. The uniaxial tensile tests were conducted at temperatures of 25 °C, 500 °C, 750 °C, 1000 °C, and 1250 °C, respectively. It was found that the three stress–strain curves recorded at five different temperatures were comparable. Typical real relationships between stress and strain are depicted in Figure 1.

#### 2.1.3. Failure Fractography

The crack samples were sectioned from the weld components by using electro-discharge machining (EDM, B50, Rixin spark machine, Shenzhen, China) followed by ultrasonic cleaning of the samples. The fracture profiles and elemental maps were then examined and probed using scanning electron microscopy (SEM, S-3700, HITACHI, Tokyo, Japan).

### 2.2. Numerical Prediction

XFEM was used to detail the evolution of the solidification crack, where the loads introducing the cracking were the outputs from the thermo-mechanical process of the welding. Therefore, the thermo-mechanical finite element method (TM-FEM) was performed first. To relate the loading condition in XFEM to the output of TM-FEM, a submodeling method was used.

#### 2.2.1. Thermo-Mechanical Modeling

The TM-FEM model was configured similarly to in the weld experiment. The weld was meshed using a finer element to account for the adequate gradients in model’s heat source. The Gaussian-heat-flux-source model was used because it is the fundamental heat-source model for performing TM-FEM analyses [21]. The thermal boundary conditions combined with the heat exchange from both convection and radiation were applied to the upper surfaces of the weld components. Moreover, different values of the combined convection–radiation coefficient were investigated until the modeled temperature field showed good accordance with the thermocouple measurements. The fitting value of the convection–radiation coefficient was calculated to be 210 W/(m^2^ K). Furthermore, constant-temperature boundary conditions were set at both weld laterals. To simulate both the movement of the heat source and the weld-bead deposition, the ABAQUS code was programmed.

After solving the governing equations of heat conductivity, the thermal results from the model were coupled into the mechanical analysis to account for the weld’s mechanical behavior in sequence. For the analysis of the welding-induced thermal work hardening, the concept of isotropic hardening is sufficient to describe the generated thermal plasticity within one tension–compression cycle. The re-strain conditions were the same as those in the welding experiment. The technique of deactivating and reactivating elements was used to simulate the weld-bead-deposition process.

#### 2.2.2. Submodeling Method

To reduce the computation cost and present the elaborate gradient around the initial defect, a local submodeling technique embedded in FEM was used. A portion of the global model was cut along the longitudinal direction at a distance far enough from free edges—satisfying Saint-Venant’s Principle—and the values for the degrees of freedom (translational, potential, etc.) and body forces (temperature) from the global model were equivalently imposed on the cut-out weld submodel.

Figure 2 depicts the location and size of the submodel to be managed, where 30,400 elements and 35,249 nodes were included. The submodel was one bead deposition extracted from the weld. The meshes in the longitudinal and transverse directions are 20 and 5 times more refined than those in the global model, respectively. The submodeling was validated by checking the stresses of nodes with identical coordinates.

#### 2.2.3. Extended Finite Element Method

In XFEM, the geometry of the crack is mesh-independent, and the mesh update is not necessary to conform to the growing crack. This is fulfilled by combining the enrichment functions with the standard finite element (FE) scheme. In the standard FE method, the displacement-vector function u(x) is approximately expressed as:(1)$$u\left(x\right)=\sum_{i=1}^{M}N_{i}\left(x\right)u_{i},$$
where $N_{i}\left(x\right)$ denotes the nodal approximation space, and $u_{i}$ the nodal displacement vector related to the continuous geometry.

According to the concept of the partition of the unity method [16,23], the presence of a crack is expressed using enrichment functions linked with additional degrees of freedom (DoFs), as depicted in Figure 3.

The enrichment functions usually divide into an asymptotic function that describes the near-tip mechanical singularity and a discontinuous function that describes the displacement jump as crossing the crack surfaces. Integrating into the standard FE displacement, the displacement-vector function $u_{\mathrm{XFEM}}$ is approximated as:(2)$$u_{\mathrm{XFEM}}\left(x\right)=\sum_{i\in S}N_{i}\left(x\right)u_{i}+\sum_{i\in S_{h}}N_{i}\left(x\right)H\left(x\right)a_{i}+\sum_{i\in S_{c}}N_{i}\left(x\right)\sum_{a=1}^{4}\varphi_{a}\left(x\right)b_{i}{}^{a},$$
where S denotes all of the nodes in model; the second term on the right-hand side of Equation (2) accounts for the product of the nodal degree of freedom vector $a_{i}$ and the corresponding enriched jump function H(x), and $S_{h}$ is the set of nodes whose shape function is divided by the interior of the crack. The third term accounts for the output of the nodal degree of freedom vector $b_{i}{}^{a}$ and the corresponding elastic asymptotic crack-tip function $\varphi_{a}\left(x\right)$; $S_{c}$ is the set of nodes whose shape function is divided by the crack tip. The jump function H(x) takes the form:(3)$$H\left(x\right)=\{\begin{array}{cc} 1 & \mathrm{if}\;\left(x-x^{*}\right)\cdot n\geq 0\\ -1 & \mathrm{otherwise} \end{array},$$
where **n** denotes the unit outwardly normal to the crack at $x^{*}$, and $x^{*}$ denotes the point on the crack closest to sample point ***x***.

In Equation (2), the elastic asymptotic crack-tip function $\varphi_{a}\left(x\right)$ can be defined using the level-set function as follows:(4)$$\varphi_{a}\left(x\right)=\left[\sqrt{r}\sin\frac{\theta }{2},\sqrt{r}\cos\frac{\theta }{2},\sqrt{r}\sin\theta \cdot \sin\frac{\theta }{2},\sqrt{r}\sin\theta \cdot \;\cos\frac{\theta }{2}\right],$$
where (*r*,θ) denotes a polar coordinate system with its origin located at the crack’s tip.

The level set method (LSM), supported by the level set function, is a powerful simulation method for capturing interface motion. The LSM is incorporated into XFEM and can be activated using signed distance function to track the crack surface and front, PHILSM and PSILSM, respectively.

Upon crack growth, the initial defect interfaces seldom penetrate each other; the “hard normal” behavior is fully covered when the solidification crack surfaces come into contact. The maximum principal stress and the energy-release-rate-based power law with a coefficient of 1 were designated as the crack-initiation criterion and the crack-evolution criterion, respectively. Furthermore, to improve the convergence behavior, the viscosity coefficient of 0.05 for the stabilization cohesive coefficient was configured.

The weld’s centerline is reported to be the initial defect’s preferred position due to low-melting-point impurities [24,25] or insufficient-fluid-feed-induced peak volumetric strain and triaxiality [26]. An initial defect was therefore centerline-positioned with dimensions of length *a* = 4 mm, height *c* = 3 mm, and embedded depth *d* = 1 mm.

#### 2.2.4. Displacement–Extrapolation Method

The SIF can be estimated using the crack-tip opening displacement (CTOD) test [27]. In terms of $K_{I}$, the SIF component can be expressed in the CTOD test as:(5)$$K_{I}=\frac{\sqrt{2\pi \;}E}{4\left(1-\upsilon^{2}\right)\sqrt{R}}V,$$
where E and V denote the modulus of elasticity and the Poisson’s ratio, respectively, and V denotes the CTOD, which can be retrieved from the XFEM output.

We suppose two points, *X* and *Y*, located on the line normal to the crack tip, with distances from the crack tip, $R_{X}$ and $R_{Y}$, respectively. Then, Equation (5) can be rewritten in terms of the displacement–extrapolation method as follows:(6)$$K_{I}=\frac{k}{k-1}K_{I,X}-\frac{1}{k-1}K_{I,Y},$$
where *k* denotes a constant with $k=\frac{R_{Y}}{R_{X}}$, and $K_{I,X}$ and $K_{I,Y}$ denote the *K*_I_ of points *X* and *Y*, respectively.

Similarly, $K_{II}$ and $K_{III}$ can also be defined.

## 3. Results

### 3.1. Modeling

At the temperature step 1196 °C (which is a typical temperature that falls in the thermal brittle-temperature range (BTR)), thermal stresses were retrieved and were imposed on the XFEM model via the submodeling method. Fracture analysis was performed and 3-D-based solutions to the crack evolution were achieved, as depicted in Figure 4.

Figure 4a depicts the evolution of the stress contour with high values at the crack front, indicating that the crack front is always the exact site of stress concentration. The variable of StatusXFEM represents the damage status varying from 0.0 to 1.0, with 0 for new surfaces and 1 for completely damaged ones, as shown in Figure 4b; the value of PHILSM of 0 denotes that the signed plane is exactly at the crack surface, as shown in Figure 4c. The upper tip *M* is the first to experience maximum principal stresses up to the point of damage. Subsequently, the crack front propagates upward, reaches the weld’s upper surface, develops along the weld’s upper surface until it has gone through the weld thickness, and finally, the lower tip *N* initiates and develops downward to the lower surface; i.e., there is a total four crack-tip failures in the sequence of M > P > N > Q.

SIFs can be readily calculated with the result of the XFEM modeling in terms of the displacement–extrapolation method listed in Equations (5) and (6). The areas of cracked surfaces and SIFs at each approximate incremental time step were calculated, as shown in Figure 5a. The SIFs show fairly good linear correlation with the cracked area, qualifying SIFs as an effective fracture parameter when linked with the driving force of solidification cracking. In fact, in the long crack case, SIFs are known to be effective [27]. This linear quantitative relationship between the cracked area and SIFs can be attributed to the dominant brittle nature of solidification cracking, and is in good agreement with the observations [21,22]. Therefore, the SIF was utilized to characterize the crack singularities in terms of the local crack-opening displacement.

The SIFs of the four crack tips are compared in Figure 5b. It shows that, of the three SIF components, the component *K*_I_ is dominant, suggesting that mode I is prevalent during the course of solidification cracking. This is because of the special orientation of the initial defect, which contributes to a large amount of strength loss and stress concentration along the longitudinal direction, enabling the weld’s transverse stress to be the dominant component among three stress components in cracking. It is also observed in Figure 5b that *K*_I, *M*_
*> K*_I, P_ and *K*_I, *N*_
*> K*_I, *Q*_, indicating that the crack’s open side is subjected to a higher susceptibility to cracking than the closed one. Additionally, at the same cross-section, the cracking shows that the upper crack tip is preferred, with *K*_I, *M*_
*> K*_I, *N*_ and *K*_I, *P*_
*> K*_I, *Q*_. This is attributed to the larger transverse stress distribution at the weld’s upper surface introduced by thermal contraction and solidification shrinkage due to the ‘V’ shape of the weld’s cross-section with the weld’s larger upper surface size.

### 3.2. Experiment

Solidification crack specimens were obtained and firstly extracted from the weldment. To get the arrested morphology of the solidification crack using the weld matrix, there was a 5 mm blank from the crack tips, as shown in Figure 6a. The appearance of the solidification crack exhibits a total longitudinal orientation with local tortuosity due to the microstructure’s sensibility [28,29]. Then, the specimen was longitudinally divided into three pairs—i.e., front, middle, and back—in accordance with the weld direction, using EDM and subsequent three-point bending tests until complete separation, as shown in Figure 6b, where the bottom three pieces were the counterparts of those placed on top. The corresponding magnifications are shown in Figure 6c–e.

Figure 6b shows three regions with typical morphologies. One is the liquid-film region, labeled as I, located between the yellow dotted line and the red dashed line, and characterized with liquid films spreading all over the grain boundaries, as magnified in Figure 6f; this is common in solidification cracking, weakening the bonding of interfaces and thus making the crack path possible under the thermal stress of the weld. The maximum embedded depth was measured to be 2.3 mm from the weld’s upper surface. The second is the open region, labeled as II in Figure 6b, with the red dashed line. This region is characterized by uneven dendritic tearing morphologies with a direction almost normal to the weld surface, as magnified in Figure 6c,d, which was attributed to the dominant open mode of crack growth driven by the force introduced by solidification shrinkage and thermal contraction. Moreover, it is interesting to see that this region concentrates at the weld’s upper surface rather than the lower surface, suggesting that the upward propagation is preferred for the solidification crack. Both the uneven morphologies and distribution are consistent with the modeling result; i.e., the solidification crack evolution in the open mode and in a preferred direction normal to weld’s upper surface. The third region is characterized by unequal dimple and shear bands—as shown in Figure 6b and the magnified fractographies in Figure 6d,e,i—which were introduced by a three-point bending test [30,31]. Farther from region I, the dimples are deeper and more non-equiaxed with a direction consistent with exerted shear stress. Additionally, an isolated block with dimple fractography at the top surface (labelled as “B3” in Figure 6d and magnified in Figure 6h) was observed. This uncracked block suggests that this solidification crack was developed interiorly from two or more separated initial defects.

It is worth noting that the boundary region of the solidification crack, labelled as B2 in Figure 6d and magnified in Figure 6g, shows two discrepancy morphologies—coarse and shallow dimples and dense and deep dimples introduced by thermal weld stress and forces in the three-point bending test, respectively. Those two morphologies even can be distinguished by a smooth curve, suggesting that the solidification crack was interiorly arrested by the weld matrix rather than propagated through the lower surface. Moreover, a small inner solidification crack was observed with an area less than 1 mm^2^ and kept close to the weld surface (Figure 6b in yellow dotted ellipse).

Metallurgical impurities are typically associated with solidification cracking. The elemental maps and compositions at different regions are shown in Figure 7 and Table 2, respectively. It can be noted that the liquid-film region was proven to have a higher oxygen content of 3.13 (wt. %) and lower Fe content of 94.08 (wt. %), compared to oxygen content 0.09 (wt. %) and Fe content 98.05 (wt. %) at the dimple region. This suggests that it was the low-strength oxidation formed during weld solidification that yielded the weld’s mushy region to the weld’s thermal stress. Similar observations can also be found in [32].

## 4. Discussion

### 4.1. Driving Force

Initial defects were usually observed to tilt to weld plate plane [13,26,33], which is expected to weaken the loading capability in both the thickness and transverse directions. The inclined angles of the initial defect susceptible to solidification cracking were investigated with the cases of 30° and 60° together with the cases of 0° and 90° for comparison, as shown in Figure 8. It can be seen that the *K*_I_ increases with the tilted angle, and the initial defect vertical to the weld plate plane yields the highest susceptibility to solidification cracking. Meanwhile, it is interesting to see that no matter the angle at which the initial defects tilt, the defects uniformly advance in a direction almost vertical to the weld plate plane, enabling the crack to appear longitudinally and centerline-oriented at the weld surface. Those predictions highlight the dominant role of the weld’s transverse stress component in driving solidification cracking rather than the stress component in the thickness direction. Similar observations are also available in [13,23] and in shipbuilding engineering. This prediction is also in agreement with the observation derived from PFM that as the grain boundary angle to the weld centerline increases, the susceptibility to solidification cracking increases [33]. The weld’s transverse stress component is therefore proven to contribute to the longitudinally centerline-oriented appearance of the solidification crack.

### 4.2. Propagation Paths

The cross-sectional propagation paths of solidification cracks are predicted to initiate from inner surfaces towards upper surfaces, as confirmed by the experimental results in Figure 6. This prediction is also in good agreement with the in situ observation of solidification cracking via high-speed, high-energy synchrotron X-ray radiography [26]. The crack propagation from the core of the weld towards the weld’s free surface was observed in the Tungsten inert gas (TIG) welding of steel. In surface observations [24], a centerline-positioned solidification crack was observed to follow the heat source until complete fracture. Similar crack evolution was also reported in Al–Cu alloy [34,35], Al–Si–Cu alloy [36], other materials [37], in contrast to another Al–Cu alloy cracking induced by pre-existing surface defects [38].

To illustrate the preferred path of solidification crack propagation, in the longitudinal direction, except for the current-positioned initial defect simulating the insufficient feeding or impurity-tracking forward electrode, there are two other scenarios of initial defect positioning: One is that the insufficient fluid feeding or impurity is terminated at the current cross-section, and thus, a closed initial defect forms; another is that after a sudden termination, insufficient liquid feeding or impurity resumes, and separated defects form [26], as shown in Figure 9a(I,III), respectively. Given those two scenarios of initial defect positioning, their cracking behaviors were modeled. For comparison, the scenario of the current-positioned initial defect is exhibited simultaneously. All scenarios assumed the same areas for the initial defects.

Figure 9 exhibits a noticeable preference of solidification crack paths. Firstly, although the cracked area occurring in the closed initial defect case shown in Figure 9b(I) is almost identical to that of the open initial defect case shown in Figure 9b(II), the *K*_I_ of the crack’s upper tip in the closed initial defect case is 4.3 MN/m^−3/2^, in contrast with 11.4 MN/m^−3/2^ in the open initial defect case, indicating that cracking is seriously localized and that the open side is preferred. This discrepancy highlights that the insufficient feeding or impurity open to the current cross-section has higher singularity than the closed one. Secondly, this preference of propagation path changes with multiple initial defects, as shown in Figure 9b(III), where two initial defects are present. The two initial defects extend alone at their first stage of growth, and then turn to converge and resultantly coalesce, instead of breaking through the weld’s upper surface as the single open defect did, as shown in case II. This behavior of the separated defects can be confirmed by the experimental observation shown in Figure 6b and similar observations in [26]. This observation highlights that the interaction of separated initial defects makes their connection region a preferable path for the crack when the two initial defects are close enough. Thirdly, from the sense of susceptibility to solidification cracking, as compared to the single larger initial defect with a smaller cracked area and lower SIF shown in Figure 9b, scattered initial defects with the same total crack areas are less prone to cracking. Similar observations can be found in [30]. Bigger initial current cracks and more concentrated weakening of cross-sectional current areas result in higher susceptibility of the current section to solidification cracking, as reported in [39,40].

## 5. Conclusions

The solidification crack profile test confirms that XFEM is an effective numerical method for reproducing weld solidification crack propagation. SIF is an effective fracture parameter to correlate with solidification cracking.Upon exposure to weld stress higher than the maximum tensile principal stress, the solidification crack initiates and inherently prefers to propagate upward towards the weld’s upper surface due to its higher weld stress.Among the three SIF components, the mode I is dominant. Initial defects open to current cross section and vertical to the weld plate plane are more susceptible to solidification cracking compared to the closed and tilted ones, respectively.The brittle nature of solidification cracking and the dominant role of the weld’s transverse stress component make the initial defect propagate vertically towards weld plate plane in spite of its initial tilting orientation. Lightly scattered initial defects show a lower total cracking susceptibility compared to highly concentrated initial defects with same area.

## Figures and Tables

**Figure 1 materials-13-00483-f001:**
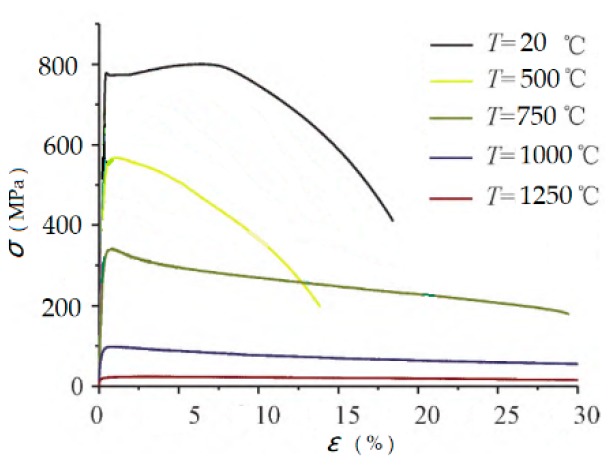
Thermo-mechanical properties of the Q690D steel.

**Figure 2 materials-13-00483-f002:**
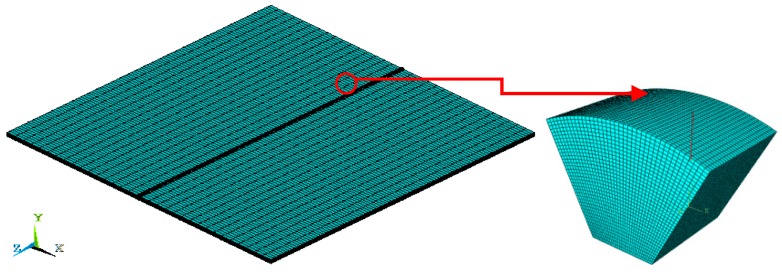
Submodeling method.

**Figure 3 materials-13-00483-f003:**
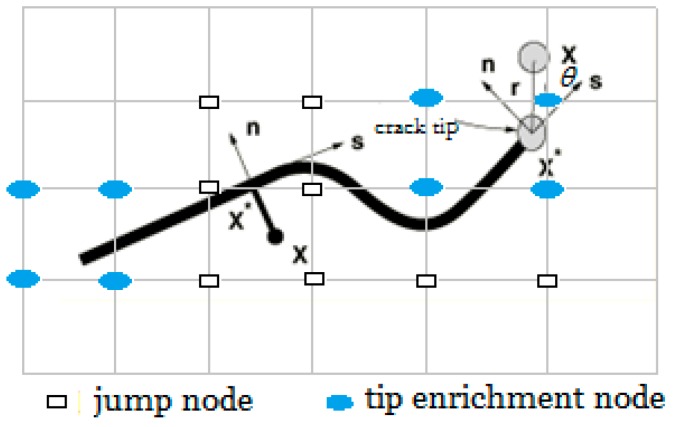
Schematic of a node in cracked enriched elements.

**Figure 4 materials-13-00483-f004:**
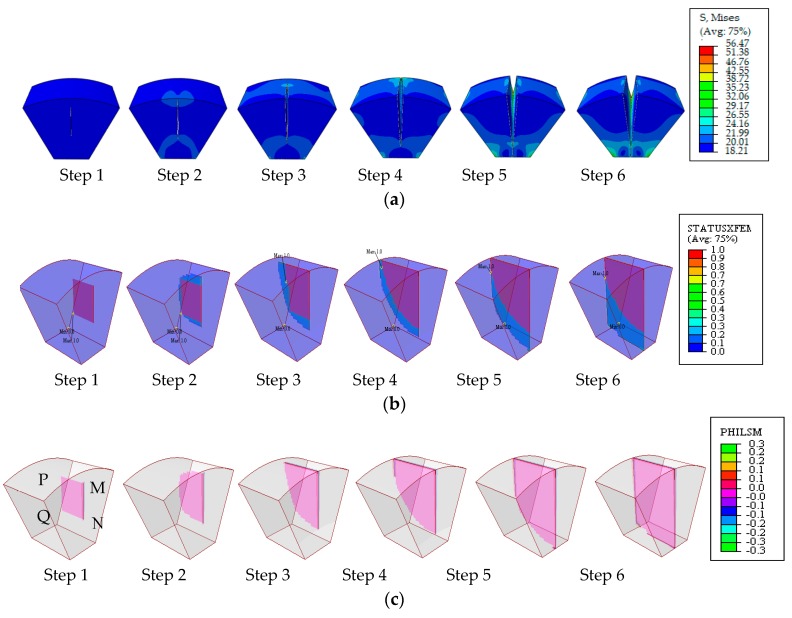
Contour plots of (**a**) Mises stress; (**b**) StatusXFEM; (**c**) PHILSM.

**Figure 5 materials-13-00483-f005:**
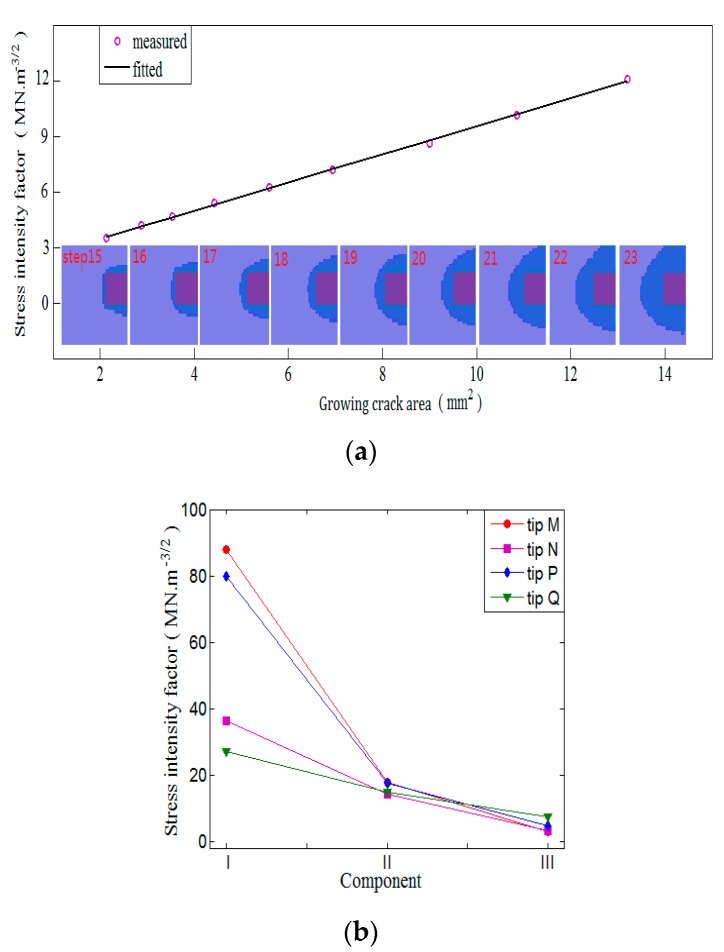
Stress intensity factor (SIF): (**a**) Validation; (**b**) comparison of crack tips.

**Figure 6 materials-13-00483-f006:**
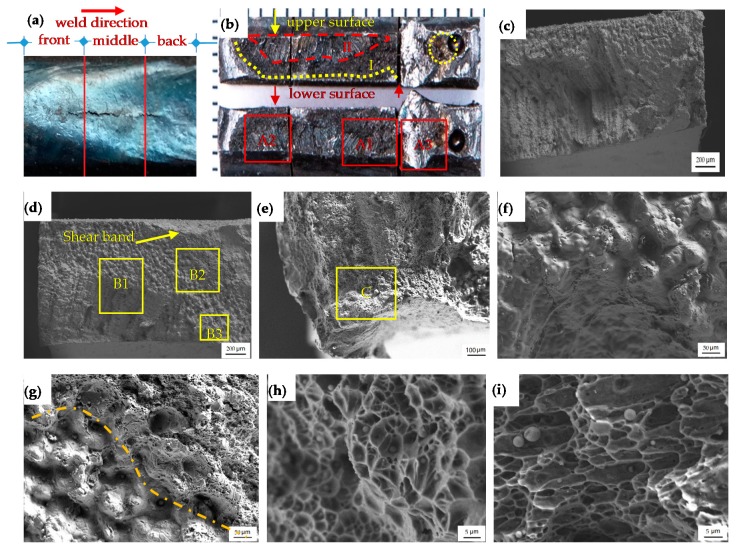
Solidification crack (**a**) appearance, (**b**) surface, and (**c**–**i**) A1, A2, A3, B1, B2, B3, and C magnifications, respectively.

**Figure 7 materials-13-00483-f007:**
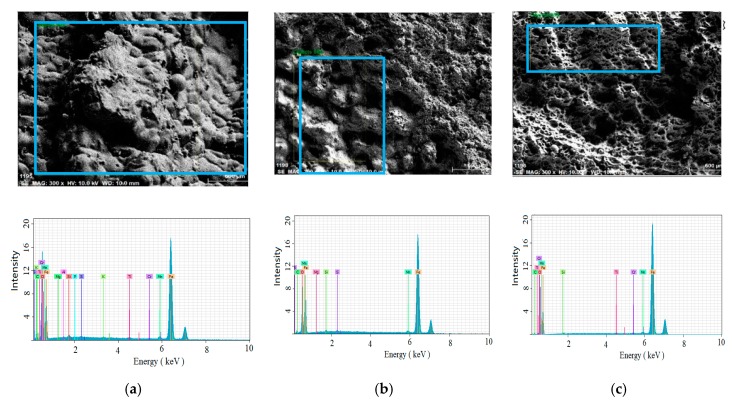
Elemental map of the (**a**) liquid-film region; (**b**) boundary region; and (**c**) dimple region.

**Figure 8 materials-13-00483-f008:**
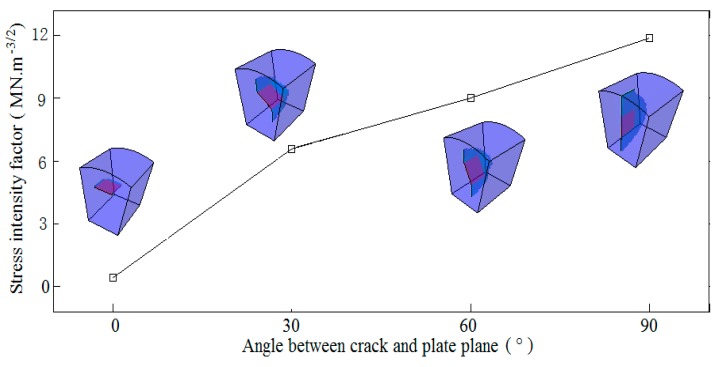
Susceptibility of the inclined angle to solidification cracking.

**Figure 9 materials-13-00483-f009:**
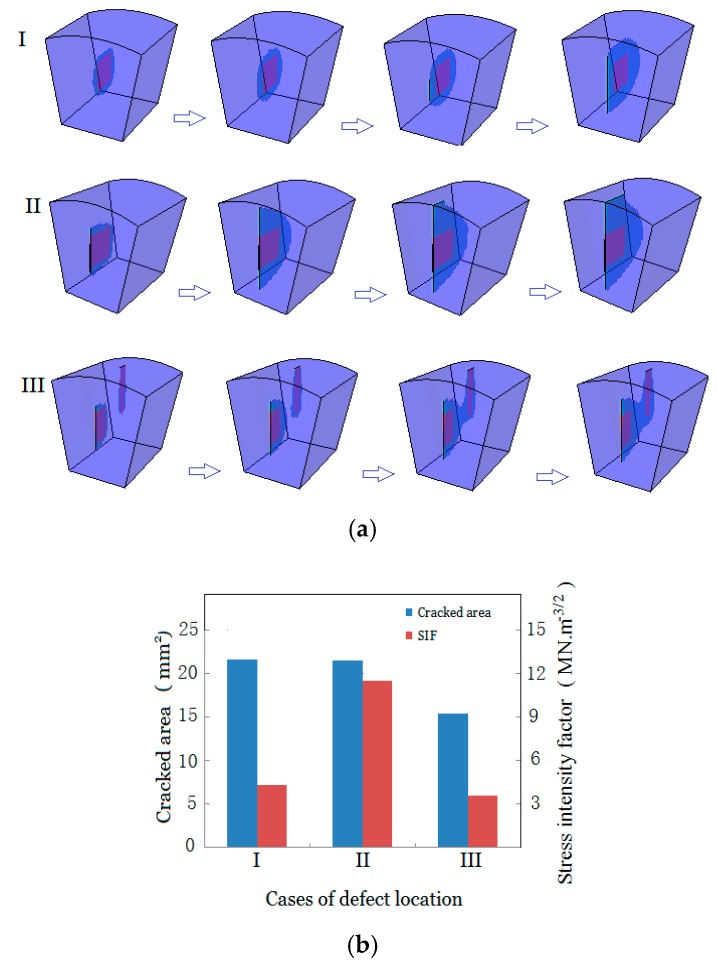
Different defect (**a**) evolutions; (**b**) susceptibilities to cracking.

**Table 1 materials-13-00483-t001:** The chemical constitution of Q690D steel (%, balance Fe).

Element	C	Si	Mn	P	S	Cr	Al	Ti	Nb	Ni	Mo
wt.	0.14	0.312	1.471	0.015	0.010	0.826	0.031	0.23	0.03	0.68	0.212

**Table 2 materials-13-00483-t002:** Elemental compositions of the crack surface at different regions (wt. %).

Element	Fe	Mn	O	C	S	Mg	Ti	Cr
Liquid-film region	94.08	1.11	3.13	1.01	-	-	0.27	0.35
Boundary region	97.75	1.06	0.80	0.39	0.01	0.0	-	-
Dimple region	98.05	0.84	0.03	0.25	-	-	0.28	0.55

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
