# Peer review of "Solidification Crack Evolution in High-Strength Steel Welding Using the Extended Finite Element Method"

_materials, 2020, doi:10.3390/ma13020483_

Round 1
Reviewer 1 Report
The current manuscript needs some corrections as followed.
Introduction(1) Authors need to provide more explanations about why high strength steels can meet the demand for light weight construction since aluminum and magnesium are common light-weight materials rather than steels (Line 27 to 31).
(2) “The increase in the strength of the weld components contributes to the appearance of solidification cracking” is confusing (Line 27 to 31).
(3) “Various modelling methods were employed and …” is not clear enough for readers (Line 32-33).
Materials and Methods(1) “TMCP” should stay after thermo-mechanical control process (Line 83).
(2) uniaxial should be replaced by Uniaxial (Line 99)
(3) What was used for chemical constitution? (Line 100)
Results(1) Figure 6 (b) is too blur to readers.
Author Response
Introduction
(1) Authors need to provide more explanations about why high strength steels can meet the demand for light weight construction since aluminum and magnesium are common light-weight materials rather than steels (Line 27 to 31).
Response: steel has comparable cost advantage as compared to the aluminum and magnesium, as illustrated in lines 27-29.
(2) “The increase in the strength of the weld components contributes to the appearance of solidification cracking” is confusing (Line 27 to 31).
Response: Factors contributed to solidification cracking including alloying degree in HSS are illustrated, see Lines 29-30.
(3) “Various modelling methods were employed and …” is not clear enough for readers (Line 32-33).
Response: This part has been rewritten to outline various numerical method modeling solidification cracking mechanical behavior in an order of TM-FEM, PFM, Continuum damage model-based method, conventional fracture-based method, and extended FEM. See lines 36-74.
Materials and Methods
(1) “TMCP” should stay after thermo-mechanical control process (Line 83).
Response: The position of “TMCP” has been modified as remarks (Line 103).
(2) “uniaxial” should be replaced by “Uniaxial” (Line 99)
Response: Replacement has been done.
(3) What was used for chemical constitution? (Line 100)
Response: It is indeed redundant in manuscript only as a link of high strength steel. Since information about element (%) at weld crack surfaces have been added in the revised version, this chemical constitution can provide comparable information after weld. Therefore, this table was taken as it is and Table 2 was added, see Lines 322-323.
Results:
Figure 6 (b) is too blur to readers.
Response: Figure 6b was replaced with a clearer one, see Line 278, Figure 6b.

Reviewer 2 Report
It is good paper about solidification crack evolution in high strength steel weld using the extended finite element method. But I have some comments.
1) I think that in the literature review it is necessary to reduce (or replace) the number of links to publications before 2000. Because this does not show the relevance of your work. In particular, you can use the attached publications on mathematical modeling and on the study of the structural-phase states of high-strength steel after welding (Konovalov, S.V., Kormyshev, V.E., Gromov, V.E., Ivanov, Y.F., Kapralov, E.V., Semin, A.P. Formation features of structure-phase states of Cr–Nb–C–V containing coatings on martensitic steel (2016) Journal of Surface Investigation, 10 (5), pp. 1119-1124. DOI: 10.1134/S1027451016050098
2) Authors made (see Figure 7) elemental map of (a) liquid-film region; (b) boundary region; and (c) dimple region. I think what need add information about elements (%) in tables. It will be more informative.
Author Response
(1) In the literature review it is necessary to reduce (or replace) the number of links to publications before 2000. Because this does not show the relevance of your work. In particular, you can use the attached publications on mathematical modeling and on the study of the structural-phase states of high-strength steel after welding (Konovalov, S.V., Kormyshev, V.E., Gromov, V.E., Ivanov, Y.F., Kapralov, E.V., Semin, A.P. Formation features of structure-phase states of Cr–Nb–C–V containing coatings on martensitic steel (2016) Journal of Surface Investigation, 10 (5), pp. 1119-1124. DOI: 10.1134/S1027451016050098
Response: Four points were considered:
(1) Previous publications 11,18 were deleted;
(2) Although publication [7] is before 2000, but as a representative of phase field method in welding solidification simulation, this publication is retained.
(3) The attached publication was referred, see lines 93-95.
(4) The Introduction Section has been rewritten, see lines 27-91.
(2) Authors made (see Figure 7) elemental map of (a) liquid-film region; (b) boundary region; and (c) dimple region. I think what need add information about elements (%) in tables. It will be more informative.
Response: The information about element (%) has been added, see lines 322-323. Corresponding illustration was revised, see lines 314-319.

Round 2
Reviewer 2 Report
No comments.